# Three-Dimensional Fracture Analysis in Functionally Graded Materials Using the Finite Block Method in Strong Form

**DOI:** 10.3390/ma16237301

**Published:** 2023-11-24

**Authors:** C. Y. Fu, Y. Yang, Y. R. Zhou, C. Z. Shi, P. H. Wen

**Affiliations:** 1Institute of Aerospace, School of Infrastructure Engineering, Nanchang University, Nanchang 330031, China; fuchengyue@email.ncu.edu.cn (C.Y.F.); yingyang@ncu.edu.cn (Y.Y.); zyr750923@163.com (Y.R.Z.); 2School of Engineering and Materials Science, Queen Mary University of London, London E1 4NS, UK

**Keywords:** finite block method, functionally graded materials, stress intensity factor, crack opening displacements

## Abstract

In this paper, the application of the strong-form finite block method (FBM) to three-dimensional fracture analysis with functionally graded materials is presented. The main idea of the strong-form FBM is that it transforms the arbitrary physical domain into a normalized domain and utilizes the direct collocation method to form a linear system. Using the mapping technique, partial differential matrices of any order can be constructed directly. Frameworks of the strong-form FBM for three-dimensional problems based on Lagrange polynomial interpolation and Chebyshev polynomial interpolation were developed. As the dominant parameters in linear elastic fracture mechanics, the stress intensity factors with functionally graded materials (FGMs) were determined according to the crack opening displacement criteria. Several numerical examples are presented using a few blocks to demonstrate the accuracy and efficiency of the strong-form FBM.

## 1. Introduction

In engineering structures such as reinforced concrete column steel beams, landing gear, and aircraft skin, the presence of cracks has a significant impact on their health. To improve the reliability of engineering structures and gain a deeper understanding of the mechanisms of cracks, many researchers have studied fracture mechanics in various fields, such as civil engineering [1,2,3], aerospace [4,5,6], and materials science engineering [7,8,9,10]. In fracture mechanics, the asymptotic stress field surrounding a crack tip is a critical research issue. And, for linear elasticity, stress intensity factors (SIFs) are essential parameters, characterizing the strength of a stress field in the vicinity of the crack tip. Also, the SIFs are the dominant terms in the William’s series expansion. Therefore, the primary object of linear fracture mechanics is to determine the corresponding SIFs. Up to now, many theoretical or numerical methods have been developed for determining SIFs [11,12,13,14,15]. However, due to the difficulty of the mathematics involved, the number of models with which SIFs can be obtained analytically is limited, especially for FGMs. With the development of computers, numerical methods have become the most commonly used to solve general crack problems. The well-known numerical methods are the mesh-dependent methods and the meshless methods.

Regarding the mesh-dependent methods, the representative methods are the finite element method (FEM) [16,17] and the boundary element method (BEM) [18,19]. The FEM discretizes the analyzed domain using small elements with different shapes, including triangular elements and quadrilateral elements for two-dimensional problems and tetrahedral elements and hexahedral elements for three-dimensional problems, granting the FEM very strong adaptability for complicate boundaries. However, for crack problems, the meshes near the crack tip have to be refined severely in order to obtain reliable results, which is usually very time consuming. Unlike the FEM, the BEM only divides the meshes on the boundary. And due to dimensional reduction and its semi-analytical character, it can obtain high accuracy while also providing timesaving properties. However, the fundamental solutions of the BEM are not easy to obtain in general problems. Furthermore, for FGMs, the boundary integral equation always needs an extra domain integral term, which robs the BEM of the advantage of dividing meshes only on the boundary. Although many methods have been proposed to transform the domain integral into a boundary integral, such as the double reciprocity method [20] and the radial integration method [21], additionally errors were introduced into the BEM. In contrast to the mesh-dependent methods, in recent decades, the meshless methods have gained much attention due to their advantage of not requiring mesh division. In meshless methods, a set of scattered nodes are distributed to discretize the physical domain and further form a linear system through corresponding kernel functions. The typically used meshless methods are the radial basis function collocation method (RBFCM) [22,23,24,25], the finite integration method (FIM) [26,27], the finite block method (FBM) [28], the method of fundamental solution (MFS) [29,30,31], the local radial basis function collocation method (LRBFCM) [32,33,34], and so on. Up to now, those meshless methods have been successfully applied in various problems, such as band structure computation in phononic crystals [35,36], elastic wave propagation [37], the simulation of water pollution [38], geometric modeling [39,40], etc.

Among the meshless methods, the FBM is a more suitable strong-form method for solving crack problems. The FBM was first proposed by Wen et al. [28] to solve elastic problems with FGMs. The main operation involved in the FBM is to transform the physical domain into the normalized domain and form the linear system by using mapping technique and a collocation method in the normalized domain, allowing the form of the node distribution in the physical domain to be neglected. And any-order partial differential matrices can be obtained by using a first-order partial differential matrix. Furthermore, due to the continuity of stresses and displacements between two adjacent blocks, the FBM can achieve higher accuracy with fewer blocks than other collocation methods. Therefore, it has attracted a great deal of research interest in recent years. Afterwards, the FBM was successfully applied to heat conduction problems [41], contact problems [42], non-linear elasticity [43], and fracture mechanics [44]. Also, many novel numerical methods have been proposed based on the idea of the FBM, such as the finite- and infinite-block Petrov–Galerkin method [45] and the element differential method [46,47,48]. The associated studies all demonstrated the superiority of the FBM. In the current paper, the strong-form FBM is further extended to analyze three-dimensional crack problems. The frameworks of the FBM based on Lagrange polynomial interpolation (FBML) and Chebyshev polynomial interpolation (FBMC) are established. The FEM solutions were selected as the benchmark and obtained using ABAQUS with the subroutine UMAT developed for FGMs. In ABAQUS, various utility subroutines are available in Abaqus/Standard, including UMAT. The UMAT user subroutine interface in ABAQUS is very powerful and efficient in numerical simulation in solid mechanics coded in Fortran and C++. The subroutine UMAT can be used to define the mechanical constitutive behavior of a material and can be used with any procedure that includes mechanical behavior. In addition, users can use solution-dependent state variables and update the stresses and solution-dependent state variables with respect to their values at the end of the increment for which they are called. All the corresponding details are described in the ABAQUS documentation [49]. The crack opening displacement (COD) criteria were used to evaluate the SIFs with FGMs.

The structure of this paper is organized as follows. In Section 2, the mapping technique for three-dimensional problems is introduced briefly. In Section 3, the frameworks of the strong-form FBM are established. In Section 4, a description of three-dimensional elasticity problems is presented. In Section 5, several numerical examples are given to demonstrate the robustness of the strong-form FBM. Some conclusions are drawn in Section 6.

## 2. Block Mapping in Three-Dimensional Problems

For three-dimensional problems, a block in a physical domain can be mapped into a normalized domain with 20 seeds, as shown in Figure 1. The shape functions are expressed as
(1)$$\begin{array}{l} N_{i}=\frac{1}{8}\left(1+\xi_{i}\xi \right)\left(1+\eta_{i}\eta \right)\left(1+\zeta_{i}\zeta \right)\left(\xi_{i}\xi +\eta_{i}\eta +\zeta_{i}\zeta -2\right),\;\mathrm{for}\;i=1,2,\ldots ,8,\\ N_{i}=\frac{1}{4}\left(1-\xi^{2}\right)\left(1+\eta_{i}\eta \right)\left(1+\zeta_{i}\zeta \right),\;\mathrm{for}\;i=9,11,17,19,\\ N_{i}=\frac{1}{4}\left(1-\eta^{2}\right)\left(1+\zeta_{i}\zeta \right)\left(1+\xi_{i}\xi \right),\;\mathrm{for}\;i=10,12,18,20,\\ N_{i}=\frac{1}{4}\left(1-\zeta^{2}\right)\left(1+\xi_{i}\xi \right)\left(1+\eta_{i}\eta \right),\;\mathrm{for}\;i=13,14,15,16, \end{array}$$

The coordinates in the physical domain can be written as
(2)$$x=\sum_{i=1}^{20}N_{i}\left(\xi ,\eta ,\zeta \right)x_{i},\;y=\sum_{i=1}^{20}N_{i}\left(\xi ,\eta ,\zeta \right)y_{i},\;z=\sum_{i=1}^{20}N_{i}\left(\xi ,\eta ,\zeta \right)z_{i},$$
in which $\left(x_{k},y_{k},z_{k}\right)$ is the coordinate of *k*th seed. The first-order partial differential operators in Cartesian coordinates of real space are
(3)$$\begin{array}{l} \frac{\partial }{\partial x}=\beta_{11}\frac{\partial }{\partial \xi }+\beta_{12}\frac{\partial }{\partial \eta }+\beta_{13}\frac{\partial }{\partial \zeta },\\ \frac{\partial }{\partial y}=\beta_{21}\frac{\partial }{\partial \xi }+\beta_{22}\frac{\partial }{\partial \eta }+\beta_{23}\frac{\partial }{\partial \zeta },\\ \frac{\partial }{\partial z}=\beta_{31}\frac{\partial }{\partial \xi }+\beta_{32}\frac{\partial }{\partial \eta }+\beta_{33}\frac{\partial }{\partial \zeta }, \end{array}$$
where the coefficients are
(4)$$\begin{array}{l} \beta_{11}=\frac{1}{J}\left(\frac{\partial y}{\partial \eta }\frac{\partial z}{\partial \varsigma }-\frac{\partial y}{\partial \varsigma }\frac{\partial z}{\partial \eta }\right),\;\beta_{12}=\frac{1}{J}\left(\frac{\partial y}{\partial \varsigma }\frac{\partial z}{\partial \xi }-\frac{\partial y}{\partial \xi }\frac{\partial z}{\partial \varsigma }\right),\;\beta_{13}=\frac{1}{J}\left(\frac{\partial y}{\partial \xi }\frac{\partial z}{\partial \eta }-\frac{\partial y}{\partial \eta }\frac{\partial z}{\partial \xi }\right),\\ \beta_{21}=\frac{1}{J}\left(\frac{\partial x}{\partial \varsigma }\frac{\partial z}{\partial \eta }-\frac{\partial x}{\partial \eta }\frac{\partial z}{\partial \varsigma }\right),\;\beta_{22}=\frac{1}{J}\left(\frac{\partial x}{\partial \xi }\frac{\partial z}{\partial \varsigma }-\frac{\partial x}{\partial \varsigma }\frac{\partial z}{\partial \xi }\right),\;\beta_{23}=\frac{1}{J}\left(\frac{\partial x}{\partial \eta }\frac{\partial z}{\partial \xi }-\frac{\partial x}{\partial \xi }\frac{\partial z}{\partial \eta }\right),\\ \beta_{31}=\frac{1}{J}\left(\frac{\partial x}{\partial \eta }\frac{\partial y}{\partial \varsigma }-\frac{\partial x}{\partial \varsigma }\frac{\partial y}{\partial \eta }\right),\;\beta_{32}=\frac{1}{J}\left(\frac{\partial x}{\partial \varsigma }\frac{\partial y}{\partial \xi }-\frac{\partial x}{\partial \xi }\frac{\partial y}{\partial \varsigma }\right),\;\beta_{33}=\frac{1}{J}\left(\frac{\partial x}{\partial \xi }\frac{\partial y}{\partial \eta }-\frac{\partial x}{\partial \eta }\frac{\partial y}{\partial \xi }\right), \end{array}$$
in which the Jacobian determinant is
(5)$$J=\frac{\partial x}{\partial \xi }\frac{\partial y}{\partial \eta }\frac{\partial z}{\partial \varsigma }-\frac{\partial x}{\partial \xi }\frac{\partial y}{\partial \varsigma }\frac{\partial z}{\partial \eta }-\frac{\partial x}{\partial \eta }\frac{\partial y}{\partial \xi }\frac{\partial z}{\partial \varsigma }+\frac{\partial x}{\partial \varsigma }\frac{\partial y}{\partial \xi }\frac{\partial z}{\partial \eta }+\frac{\partial x}{\partial \eta }\frac{\partial y}{\partial \varsigma }\frac{\partial z}{\partial \xi }-\frac{\partial x}{\partial \varsigma }\frac{\partial y}{\partial \eta }\frac{\partial z}{\partial \xi }.$$

From Equation (3), the second-order partial differential operators can be obtained by using the derivative rule, as follows:(6)$$\begin{array}{l} \frac{\partial^{2}}{\partial x^{2}}=\beta_{11}{}^{2}\frac{\partial^{2}}{\partial \xi^{2}}+\beta_{12}{}^{2}\frac{\partial^{2}}{\partial \eta^{2}}+\beta_{13}{}^{2}\frac{\partial^{2}}{\partial \varsigma^{2}}+2\beta_{11}\beta_{12}\frac{\partial^{2}}{\partial \xi \partial \eta }+2\beta_{11}\beta_{13}\frac{\partial^{2}}{\partial \xi \partial \varsigma }+2\beta_{12}\beta_{13}\frac{\partial^{2}}{\partial \eta \partial \varsigma }\\ \;\;+\left[\beta_{11}\frac{\partial \beta_{11}}{\partial \xi }+\beta_{12}\frac{\partial \beta_{11}}{\partial \eta }+\beta_{13}\frac{\partial \beta_{11}}{\partial \varsigma }\right]\frac{\partial }{\partial \xi }+\left[\beta_{11}\frac{\partial \beta_{12}}{\partial \xi }+\beta_{12}\frac{\partial \beta_{12}}{\partial \eta }+\beta_{13}\frac{\partial \beta_{12}}{\partial \varsigma }\right]\frac{\partial }{\partial \eta }\\ \;\;+\left[\beta_{11}\frac{\partial \beta_{13}}{\partial \xi }+\beta_{12}\frac{\partial \beta_{13}}{\partial \eta }+\beta_{13}\frac{\partial \beta_{13}}{\partial \varsigma }\right]\frac{\partial }{\partial \varsigma }, \end{array}$$
(7)$$\begin{array}{l} \frac{\partial^{2}}{\partial y^{2}}=\beta_{21}{}^{2}\frac{\partial^{2}}{\partial \xi^{2}}+\beta_{22}{}^{2}\frac{\partial^{2}}{\partial \eta^{2}}+\beta_{23}{}^{2}\frac{\partial^{2}}{\partial \varsigma^{2}}+2\beta_{21}\beta_{22}\frac{\partial^{2}}{\partial \xi \partial \eta }+2\beta_{21}\beta_{23}\frac{\partial^{2}}{\partial \xi \partial \varsigma }+2\beta_{22}\beta_{23}\frac{\partial^{2}}{\partial \eta \partial \varsigma }\\ \;\;+\left[\beta_{21}\frac{\partial \beta_{21}}{\partial \xi }+\beta_{22}\frac{\partial \beta_{21}}{\partial \eta }+\beta_{23}\frac{\partial \beta_{21}}{\partial \varsigma }\right]\frac{\partial }{\partial \xi }+\left[\beta_{21}\frac{\partial \beta_{22}}{\partial \xi }+\beta_{22}\frac{\partial \beta_{22}}{\partial \eta }+\beta_{23}\frac{\partial \beta_{22}}{\partial \varsigma }\right]\frac{\partial }{\partial \eta }\\ \;\;+\left[\beta_{21}\frac{\partial \beta_{23}}{\partial \xi }+\beta_{22}\frac{\partial \beta_{23}}{\partial \eta }+\beta_{23}\frac{\partial \beta_{23}}{\partial \varsigma }\right]\frac{\partial }{\partial \varsigma }, \end{array}$$
(8)$$\begin{array}{l} \frac{\partial^{2}}{\partial z^{2}}=\beta_{31}{}^{2}\frac{\partial^{2}}{\partial \xi^{2}}+\beta_{32}{}^{2}\frac{\partial^{2}}{\partial \eta^{2}}+\beta_{33}{}^{2}\frac{\partial^{2}}{\partial \varsigma^{2}}+2\beta_{31}\beta_{32}\frac{\partial^{2}}{\partial \xi \partial \eta }+2\beta_{31}\beta_{33}\frac{\partial^{2}}{\partial \xi \partial \varsigma }+2\beta_{32}\beta_{33}\frac{\partial^{2}}{\partial \eta \partial \varsigma }\\ \;\;+\left[\beta_{31}\frac{\partial \beta_{31}}{\partial \xi }+\beta_{32}\frac{\partial \beta_{31}}{\partial \eta }+\beta_{33}\frac{\partial \beta_{31}}{\partial \varsigma }\right]\frac{\partial }{\partial \xi }+\left[\beta_{31}\frac{\partial \beta_{32}}{\partial \xi }+\beta_{32}\frac{\partial \beta_{32}}{\partial \eta }+\beta_{33}\frac{\partial \beta_{32}}{\partial \varsigma }\right]\frac{\partial }{\partial \eta }\\ \;\;+\left[\beta_{31}\frac{\partial \beta_{33}}{\partial \xi }+\beta_{32}\frac{\partial \beta_{33}}{\partial \eta }+\beta_{33}\frac{\partial \beta_{33}}{\partial \varsigma }\right]\frac{\partial }{\partial \varsigma }, \end{array}$$
(9)$$\begin{array}{l} \frac{\partial^{2}}{\partial x\partial y}=\beta_{11}\beta_{21}\frac{\partial^{2}}{\partial \xi^{2}}+\beta_{12}\beta_{22}\frac{\partial^{2}}{\partial \eta^{2}}+\beta_{13}\beta_{23}\frac{\partial^{2}}{\partial \varsigma^{2}}+\left(\beta_{11}\beta_{22}+\beta_{12}\beta_{21}\right)\frac{\partial^{2}}{\partial \xi \partial \eta }+\left(\beta_{11}\beta_{23}+\beta_{13}\beta_{21}\right)\frac{\partial^{2}}{\partial \xi \partial \varsigma }\\ \;\;+\left(\beta_{12}\beta_{23}+\beta_{13}\beta_{22}\right)\frac{\partial^{2}}{\partial \eta \partial \varsigma }+\left[\beta_{21}\frac{\partial \beta_{11}}{\partial \xi }+\beta_{22}\frac{\partial \beta_{11}}{\partial \eta }+\beta_{23}\frac{\partial \beta_{11}}{\partial \varsigma }\right]\frac{\partial }{\partial \xi }\\ \;\;+\left[\beta_{21}\frac{\partial \beta_{12}}{\partial \xi }+\beta_{22}\frac{\partial \beta_{12}}{\partial \eta }+\beta_{23}\frac{\partial \beta_{12}}{\partial \varsigma }\right]\frac{\partial }{\partial \eta }+\left[\beta_{21}\frac{\partial \beta_{13}}{\partial \xi }+\beta_{22}\frac{\partial \beta_{13}}{\partial \eta }+\beta_{23}\frac{\partial \beta_{13}}{\partial \varsigma }\right]\frac{\partial }{\partial \varsigma }, \end{array}$$
(10)$$\begin{array}{l} \frac{\partial^{2}}{\partial y\partial z}=\beta_{21}\beta_{31}\frac{\partial^{2}}{\partial \xi^{2}}+\beta_{22}\beta_{32}\frac{\partial^{2}}{\partial \eta^{2}}+\beta_{23}\beta_{33}\frac{\partial^{2}}{\partial \varsigma^{2}}+\left(\beta_{21}\beta_{32}+\beta_{22}\beta_{31}\right)\frac{\partial^{2}}{\partial \xi \partial \eta }\\ +\left(\beta_{21}\beta_{33}+\beta_{23}\beta_{31}\right)\frac{\partial^{2}}{\partial \xi \partial \varsigma }+\left(\beta_{22}\beta_{33}+\beta_{23}\beta_{32}\right)\frac{\partial^{2}}{\partial \eta \partial \varsigma }\\ +\left[\beta_{31}\frac{\partial \beta_{21}}{\partial \xi }+\beta_{32}\frac{\partial \beta_{21}}{\partial \eta }+\beta_{33}\frac{\partial \beta_{21}}{\partial \varsigma }\right]\frac{\partial }{\partial \xi }+\left[\beta_{31}\frac{\partial \beta_{22}}{\partial \xi }+\beta_{32}\frac{\partial \beta_{22}}{\partial \eta }+\beta_{33}\frac{\partial \beta_{22}}{\partial \varsigma }\right]\frac{\partial }{\partial \eta }\\ +\left[\beta_{31}\frac{\partial \beta_{23}}{\partial \xi }+\beta_{32}\frac{\partial \beta_{23}}{\partial \eta }+\beta_{33}\frac{\partial \beta_{23}}{\partial \varsigma }\right]\frac{\partial }{\partial \varsigma }, \end{array}$$
(11)$$\begin{array}{l} \frac{\partial^{2}}{\partial z\partial x}=\beta_{11}\beta_{31}\frac{\partial^{2}}{\partial \xi^{2}}+\beta_{12}\beta_{32}\frac{\partial^{2}}{\partial \eta^{2}}+\beta_{13}\beta_{33}\frac{\partial^{2}}{\partial \varsigma^{2}}+\left(\beta_{11}\beta_{32}+\beta_{12}\beta_{31}\right)\frac{\partial^{2}}{\partial \xi \partial \eta }\\ +\left(\beta_{11}\beta_{33}+\beta_{13}\beta_{31}\right)\frac{\partial^{2}}{\partial \xi \partial \varsigma }+\left(\beta_{12}\beta_{33}+\beta_{13}\beta_{32}\right)\frac{\partial^{2}}{\partial \eta \partial \varsigma }\\ +\left[\beta_{31}\frac{\partial \beta_{11}}{\partial \xi }+\beta_{32}\frac{\partial \beta_{11}}{\partial \eta }+\beta_{33}\frac{\partial \beta_{11}}{\partial \varsigma }\right]\frac{\partial }{\partial \xi }+\left[\beta_{31}\frac{\partial \beta_{12}}{\partial \xi }+\beta_{32}\frac{\partial \beta_{12}}{\partial \eta }+\beta_{33}\frac{\partial \beta_{12}}{\partial \varsigma }\right]\frac{\partial }{\partial \eta }\\ +\left[\beta_{31}\frac{\partial \beta_{13}}{\partial \xi }+\beta_{32}\frac{\partial \beta_{13}}{\partial \eta }+\beta_{33}\frac{\partial \beta_{13}}{\partial \varsigma }\right]\frac{\partial }{\partial \varsigma }. \end{array}$$

## 3. Strong-Form Finite Block Method

The strong-form numerical method is also called the collocation method. In the strong-form finite block method, the direct collocation method is used to form a differential matrix in the normalized domain. Using the mapping technique, a differential matrix in the physical domain can then be constructed. In the following section, the approximation of the strong-form finite block method is introduced.

### 3.1. A Brief Description of the FBML

In the normalized domain shown in Figure 1, the smooth function u(ξ,η,ζ) can be approximated using Lagrange polynomial as
(12)$$u\left(\xi ,\eta ,\zeta \right)=\sum_{i=1}^{N}\sum_{j=1}^{N}\sum_{k=1}^{N}F\left(\xi ,\xi_{i}\right)G\left(\eta ,\eta_{j}\right)H\left(\zeta ,\zeta_{k}\right)u_{q}=\sum_{q=1}^{N^{3}}\Phi \left(\xi ,\eta ,\zeta \right)u_{q},$$
in which Φ(ξ,η,ζ) is the shape function, *N* is the number of collocation nodes along axis *x* (*y* or *z*), and $q=\left(k-1\right)N^{2}+(j-1)N+i$, $u_{q}$ denotes the nodal value. The collocation nodes $\xi_{i}$, $\eta_{j}$, and $\zeta_{k}$ are selected as uniform nodes
(13)$$w_{p}=-1+\frac{2\left(p-1\right)}{N-1},\;p=1,2,\ldots ,N,\;w=\xi ,\eta ,\zeta ,\;p=i,j,k,$$
or roots of the Chebyshev polynomial of the first kind
(14)$$w_{p}=-\cos\frac{\pi \left(p-1\right)}{N-1},\;p=1,2,\ldots ,N,\;w=\xi ,\eta ,\zeta ,\;p=i,j,k,$$
and the polynomial functions are defined as follows:(15)$$F\left(\xi ,\xi_{i}\right)=\prod_{\begin{array}{l} l=1\\ l\neq i \end{array}}^{N}\frac{\left(\xi -\xi_{l}\right)}{\left(\xi_{i}-\xi_{l}\right)},\;G\left(\eta ,\eta_{j}\right)=\prod_{\begin{array}{l} m=1\\ m\neq j \end{array}}^{N}\frac{\left(\eta -\eta_{m}\right)}{\left(\eta_{j}-\eta_{m}\right)},\;H\left(\zeta ,\zeta_{k}\right)=\prod_{\begin{array}{l} n=1\\ n\neq j \end{array}}^{N}\frac{\left(\zeta -\zeta_{n}\right)}{\left(\zeta_{k}-\zeta_{n}\right)}.$$

Then, the first-order partial derivatives of shape function Φ with respect to ξ, η, and ζ can be determined straightforwardly as follows
(16)$$\frac{\partial \Phi }{\partial \xi }=\frac{\partial F\left(\xi ,\xi_{i}\right)}{\partial \xi }G\left(\eta ,\eta_{j}\right)H\left(\zeta ,\zeta_{k}\right),$$
(17)$$\frac{\partial \Phi }{\partial \eta }=F\left(\xi ,\xi_{i}\right)\frac{\partial G\left(\eta ,\eta_{j}\right)}{\partial \eta }H\left(\zeta ,\zeta_{k}\right),$$
(18)$$\frac{\partial \Phi }{\partial \zeta }=F\left(\xi ,\xi_{i}\right)G\left(\eta ,\eta_{j}\right)\frac{\partial H\left(\zeta ,\zeta_{k}\right)}{\partial \zeta },$$
in which
(19)$$\frac{\partial F\left(\xi ,\xi_{i}\right)}{\partial \xi }=\prod_{\begin{array}{l} l=1\\ l\neq i \end{array}}^{N}{\left(\xi_{i}-\xi_{l}\right)}^{-1}\sum_{\begin{array}{l} h=1\\ h\neq i \end{array}}^{N}\prod_{\begin{array}{l} l=1\\ l\neq i\\ l\neq h \end{array}}^{N}\left(\xi -\xi_{l}\right),$$
(20)$$\frac{\partial G\left(\eta ,\eta_{j}\right)}{\partial \eta }=\prod_{\begin{array}{l} m=1\\ m\neq j \end{array}}^{N}{\left(\eta_{j}-\eta_{m}\right)}^{-1}\sum_{\begin{array}{l} h=1\\ h\neq j \end{array}}^{N}\prod_{\begin{array}{l} m=1\\ m\neq j\\ m\neq h \end{array}}^{N}\left(\eta -\eta_{m}\right),$$
(21)$$\frac{\partial H\left(\zeta ,\zeta_{k}\right)}{\partial \zeta }=\prod_{\begin{array}{l} n=1\\ n\neq j \end{array}}^{N}{\left(\zeta_{j}-\zeta_{k}\right)}^{-1}\sum_{\begin{array}{l} h=1\\ h\neq k \end{array}}^{N}\prod_{\begin{array}{l} n=1\\ n\neq k\\ n\neq h \end{array}}^{N}\left(\zeta -\zeta_{n}\right).$$

Substituting all the collocation nodes into partial derivatives of the shape function, the matrix form of the first-order partial derivatives of *u* can be formulated as
(22)$$U_{\xi }=D_{\xi }u,\;U_{\eta }=D_{\eta }u,\;U_{\zeta }=D_{\zeta }u,$$
in which $u=\left[u_{1},u_{2},\ldots ,u_{N^{3}}\right]$ is the vector of nodal values with sizes of $N^{3}\times 1$, while $D_{\xi }$, $D_{\eta }$, and $D_{\zeta }$ are all the first-order differential matrices with dimensions of $N^{3}\times N^{3}$. Thus, we have
(23)$$D_{\xi }=\left[\begin{array}{l} D_{0}\;0\;\cdots \;0\\ 0\;D_{0}\;\cdots \;0\\ \cdots \;\cdots \;\;\;\;\cdots \;\;\;\cdots \\ 0\;0\;\cdots \;D_{0} \end{array}\right],$$
where $D_{0}$ is the first-order differential matrix for a one-dimensional case with dimensions of N×N, which can be obtained from Equation (19). By using the transformation matrix $T_{\eta }$ and $T_{\zeta }$, one can obtain
(24)$$D_{\eta }=T_{\eta }D_{\xi }T_{\eta }{}^{-1},\;D_{\zeta }=T_{\zeta }D_{\xi }T_{\zeta }{}^{-1},$$
in which
(25)$$T^{\eta }{}_{\left(k-1\right)N^{2}+\left(j-1\right)N+i,\;\left(k-1\right)N^{2}+\left(i-1\right)N+j}=1,\;i,j,k=1,2,\ldots ,N,$$
(26)$$T^{\zeta }{}_{\left(k-1\right)N^{2}+\left(j-1\right)N+i,\;\left(i-1\right)N^{2}+\left(j-1\right)N+k}=1,\;i,j,k=1,2,\ldots ,N.$$

For any-order partial derivatives with respect to ξ, η, and ζ, we have
(27)$$U_{\xi \eta \zeta }^{\left(d_{1}d_{2}d_{3}\right)}=D_{\xi }^{d_{1}}D_{\eta }^{d_{2}}D_{\zeta }^{d_{3}}u,\;d_{1},d_{2},d_{3}\geq 0.$$

Therefore, from Equation (3), one can obtain
(28)$$U_{x}=B_{11}U_{\xi }+B_{12}U_{\eta }+B_{13}U_{\zeta }=\left(B_{11}D_{\xi }+B_{12}D_{\eta }+B_{13}D_{\zeta }\right)u=D_{x}u,$$
(29)$$U_{y}=B_{21}U_{\xi }+B_{22}U_{\eta }+B_{23}U_{\zeta }=\left(B_{21}D_{\xi }+B_{22}D_{\eta }+B_{23}D_{\zeta }\right)u=D_{y}u,$$
(30)$$U_{z}=B_{31}U_{\xi }+B_{32}U_{\eta }+B_{33}U_{\zeta }=\left(B_{31}D_{\xi }+B_{32}D_{\eta }+B_{33}D_{\zeta }\right)u=D_{z}u,$$
where $D_{x}$, $D_{y}$, and $D_{z}$ constitute the first-order differential matrix in the physical domain, and $B_{\gamma \mu }$ denotes the diagonal matrix, defined as
(31)$$B_{\gamma \mu }=\left[\begin{array}{l} \beta_{\gamma \mu }^{\left(1\right)}\;\;0\;\;\;\;\cdots \;\;0\\ \;\;0\;\;\;\beta_{\gamma \mu }^{\left(2\right)}\;\;\;\;\cdots \;\;0\\ \;\cdots \;\;\;\;\cdots \;\;\;\;\cdots \;\;\;\;\cdots \\ \;\;0\;\;0\;\;\;\cdots \;\;\;\;\beta_{\gamma \mu }^{\left(N^{3}\right)} \end{array}\right],\;\gamma ,\mu =1,2,3.$$

Similarly, any-order partial derivatives in the physical domain with respect to x, y, and z can be expressed as
(32)$$U_{xyz}^{\left(d_{1}d_{2}d_{3}\right)}=D_{x}^{d_{1}}D_{y}^{d_{2}}D_{z}^{d_{3}}u,\;d_{1},d_{2},d_{3}\geq 0.$$

### 3.2. The Brief of the FBMC

Besides Lagrange polynomial interpolation, the framework of the FBM can also be constructed using Chebyshev polynomial interpolation. The unknown function u(ξ,η,ζ) can be written in the following form
(33)$$u\left(\xi ,\eta ,\zeta \right)=\sum_{i=1}^{N}\sum_{j=1}^{N}\sum_{k=1}^{N}\alpha_{ijk}T_{i}\left(\xi \right)T_{j}\left(\eta \right)T_{k}\left(\zeta \right),$$
where $\alpha_{ijk}$ is the unknown coefficient and $T_{i}\left(\xi \right)$, $T_{j}\left(\eta \right)$, and $T_{k}\left(\zeta \right)$ are *i*-th, *j*-th, and *k*-th order Chebyshev polynomials of the first kind, respectively. According to the properties of the Chebyshev polynomial, the first-order and second-order partial derivatives of *u* with respect to ξ, η, and ζ can be obtained analytically as follows
(34)$$\frac{\partial u}{\partial \xi }=\sum_{i=1}^{N}\sum_{j=1}^{N}\sum_{k=1}^{N}\alpha_{ijk}jU_{i-1}\left(\xi \right)T_{i}\left(\eta \right)T_{k}\left(\zeta \right),$$
(35)$$\frac{\partial u}{\partial \eta }=\sum_{i=1}^{N}\sum_{j=1}^{N}\sum_{k=1}^{N}\alpha_{ijk}jU_{j-1}\left(\eta \right)T_{i}\left(\xi \right)T_{k}\left(\zeta \right),$$
(36)$$\frac{\partial u}{\partial \zeta }=\sum_{i=1}^{N}\sum_{j=1}^{N}\sum_{k=1}^{N}\alpha_{ijk}kU_{k-1}\left(\zeta \right)T_{i}\left(\xi \right)T_{k}\left(\eta \right),$$
(37)$$\frac{\partial^{2}u}{\partial \xi^{2}}=\sum_{i=1}^{N}\sum_{j=1}^{N}\sum_{k=1}^{N}\alpha_{ijk}i\frac{\left(i+1\right)T_{i}\left(\xi \right)-U_{i}\left(\xi \right)}{\xi^{2}-1}T_{j}\left(\eta \right)T_{k}\left(\zeta \right),$$
(38)$$\frac{\partial^{2}u}{\partial \eta^{2}}=\sum_{i=1}^{N}\sum_{j=1}^{N}\sum_{k=1}^{N}\alpha_{ijk}j\frac{\left(j+1\right)T_{j}\left(\eta \right)-U_{j}\left(\eta \right)}{\eta^{2}-1}T_{i}\left(\xi \right)T_{k}\left(\zeta \right),$$
(39)$$\frac{\partial^{2}u}{\partial \zeta^{2}}=\sum_{i=1}^{N}\sum_{j=1}^{N}\sum_{k=1}^{N}\alpha_{ijk}k\frac{\left(k+1\right)T_{k}\left(\zeta \right)-U_{k}\left(\zeta \right)}{\zeta^{2}-1}T_{i}\left(\xi \right)T_{j}\left(\eta \right),$$
(40)$$\frac{\partial^{2}u}{\partial \xi \partial \eta }=\sum_{i=1}^{N}\sum_{j=1}^{N}\sum_{k=1}^{N}\alpha_{ijk}ijU_{i-1}\left(\xi \right)U_{j-1}\left(\eta \right)T_{k}\left(\zeta \right),$$
(41)$$\frac{\partial^{2}u}{\partial \eta \partial \zeta }=\sum_{i=1}^{N}\sum_{j=1}^{N}\sum_{k=1}^{N}\alpha_{ijk}jkU_{j-1}\left(\eta \right)U_{k-1}\left(\zeta \right)T_{i}\left(\xi \right),$$
(42)$$\frac{\partial^{2}u}{\partial \zeta \partial \xi }=\sum_{i=1}^{N}\sum_{j=1}^{N}\sum_{k=1}^{N}\alpha_{ijk}kiU_{k-1}\left(\xi \right)U_{i-1}\left(\zeta \right)T_{j}\left(\eta \right),$$
in which $U_{p}\left(w\right)$, w=ξ,η,ζ, p=i,j,k, denotes the *p*-th order Chebyshev polynomial of the second kind. Similar to the Lagrange polynomial interpolation, by substituting all uniform nodes or roots of the Chebyshev polynomial of the first kind into Equations (34)–(42), the matrix form of different orders of partial derivatives is expressed as
(43)$$U_{\xi }=D_{\xi }\alpha ,\;U_{\eta }=D_{\eta }\alpha ,\;U_{\zeta }=D_{\zeta }\alpha ,$$
(44)$$U_{\xi \xi }=D_{\xi \xi }\alpha ,\;U_{\eta \eta }=D_{\eta \eta }\alpha ,\;U_{\zeta \zeta }=D_{\zeta \zeta }\alpha ,$$
(45)$$U_{\xi \eta }=D_{\xi \eta }\alpha ,\;U_{\eta \zeta }=D_{\eta \zeta }\alpha ,\;U_{\zeta \xi }=D_{\zeta \xi }\alpha ,$$
where $\alpha ={\left[\alpha_{1},\alpha_{2},\ldots ,\alpha_{N^{3}}\right]}^{T}$ is the vector of unknown coefficients with dimensions of $N^{3}\times 1$, while $D_{\kappa }$ and $D_{\kappa \chi }$, for which κ,χ=ξ,η,ζ, are differential matrices in the normalized domain with dimensions of $N^{3}\times N^{3}$. From Equations (3)–(11), the matrix form of the first-order partial derivatives in the physical domain can be formulated as
(46)$$U_{x}=\left(B_{11}D_{\xi }+B_{12}D_{\eta }+B_{13}D_{\zeta }\right)\alpha =\Theta_{x}\alpha ,$$
(47)$$U_{y}=\left(B_{21}D_{\xi }+B_{22}D_{\eta }+B_{33}D_{\zeta }\right)\alpha =\Theta_{y}\alpha ,$$
(48)$$U_{z}=\left(B_{31}D_{\xi }+B_{32}D_{\eta }+B_{33}D_{\zeta }\right)\alpha =\Theta_{z}\alpha ,$$ in which $\Theta_{x}$, $\Theta_{y}$, and $\Theta_{z}$ are the first-order differential matrices. For second-order partial derivatives, we have
(49)$$U_{bd}=\left(\begin{array}{l} C_{1}^{\left(bd\right)}D_{\xi \xi }+C_{2}^{\left(bd\right)}D_{\eta \eta }+C_{3}^{\left(bd\right)}D_{\zeta \zeta }\\ +C_{4}^{\left(bd\right)}D_{\xi \eta }+C_{5}^{\left(bd\right)}D_{\xi \zeta }+C_{6}^{\left(bd\right)}D_{\eta \zeta }\\ +C_{7}^{\left(bd\right)}D_{\xi }+C_{8}^{\left(bd\right)}D_{\eta }+C_{9}^{\left(bd\right)}D_{\zeta } \end{array}\right)\alpha =\Theta_{bd}\alpha ,\;b,d=x,y,z,$$
in which the coefficient matrices $C_{t}^{\left(bd\right)}$, t=1,2,…,9, are presented in Appendix A.

## 4. Three-Dimensional Elasticity

### 4.1. Governing Equations

For 3D static elasticity, the governing equations of the FGMs are given as
(50)$$\frac{\partial \sigma_{xx}}{\partial x}+\frac{\partial \sigma_{xy}}{\partial y}+\frac{\partial \sigma_{xz}}{\partial z}+f_{x}=0,$$
(51)$$\frac{\partial \sigma_{xy}}{\partial x}+\frac{\partial \sigma_{yy}}{\partial y}+\frac{\partial \sigma_{yz}}{\partial z}+f_{y}=0,$$
(52)$$\frac{\partial \sigma_{xz}}{\partial x}+\frac{\partial \sigma_{yz}}{\partial y}+\frac{\partial \sigma_{zz}}{\partial z}+f_{z}=0,$$
where $\sigma_{xx}$, $\sigma_{yy}$, $\sigma_{zz}$, $\sigma_{xy}$, $\sigma_{xz}$, and $\sigma_{yz}$ denote stress components, while $f_{x}$, $f_{y}$, and $f_{z}$ are the body force components, respectively. The stress components are defined in the following way
(53)$$\sigma_{xx}=\left(\lambda +2G\right)\frac{\partial u_{x}}{\partial x}+\lambda \frac{\partial u_{y}}{\partial y}+\lambda \frac{\partial u_{z}}{\partial z},$$
(54)$$\sigma_{yy}=\lambda \frac{\partial u_{x}}{\partial x}+\left(\lambda +2G\right)\frac{\partial u_{y}}{\partial y}+\lambda \frac{\partial u_{z}}{\partial z},$$
(55)$$\sigma_{zz}=\lambda \frac{\partial u_{x}}{\partial x}+\lambda \frac{\partial u_{y}}{\partial y}+\left(\lambda +2G\right)\frac{\partial u_{z}}{\partial z},$$
(56)$$\sigma_{xy}=G\left(\frac{\partial u_{y}}{\partial x}+\frac{\partial u_{x}}{\partial y}\right),\;\sigma_{xz}=G\left(\frac{\partial u_{x}}{\partial z}+\frac{\partial u_{z}}{\partial x}\right),\;\sigma_{yz}=G\left(\frac{\partial u_{z}}{\partial y}+\frac{\partial u_{y}}{\partial z}\right),$$
in which $u_{x}$, $u_{y}$, and $u_{z}$ indicate the displacement components; λ=Eν/(1+ν)/(1−2ν), G=E/2/(1+ν), and E=E(x,y,z) denote the Young’s modulus with material gradation; and ν indicates the Poisson’s ratio. The governing equation can be further formulated as
(57)$$\begin{array}{l} k_{3}\frac{\partial^{2}u_{y}}{\partial x\partial y}+G_{y}\frac{\partial u_{y}}{\partial x}+\lambda_{x}\frac{\partial u_{y}}{\partial y}+k_{3}\frac{\partial^{2}u_{z}}{\partial x\partial z}+G_{z}\frac{\partial u_{z}}{\partial x}+\lambda_{x}\frac{\partial u_{z}}{\partial z}\\ +k_{1}\frac{\partial^{2}u_{x}}{\partial x^{2}}+k_{2}\frac{\partial^{2}u_{x}}{\partial y^{2}}+k_{2}\frac{\partial^{2}u_{x}}{\partial z^{2}}+\left(\lambda_{x}+2G_{x}\right)\frac{\partial u_{x}}{\partial x}+G_{y}\frac{\partial u_{x}}{\partial y}+G_{z}\frac{\partial u_{x}}{\partial z}+f_{x}=0, \end{array}$$
(58)$$\begin{array}{l} k_{3}\frac{\partial^{2}u_{x}}{\partial x\partial y}+\lambda_{y}\frac{\partial u_{x}}{\partial x}+G_{x}\frac{\partial u_{x}}{\partial y}+k_{3}\frac{\partial^{2}u_{z}}{\partial y\partial z}+G_{z}\frac{\partial u_{z}}{\partial y}+\lambda_{y}\frac{\partial u_{z}}{\partial z}+\\ +k_{2}\frac{\partial^{2}u_{y}}{\partial x^{2}}+k_{1}\frac{\partial^{2}u_{y}}{\partial y^{2}}+k_{2}\frac{\partial^{2}u_{y}}{\partial z^{2}}+G_{x}\frac{\partial u_{y}}{\partial x}+\left(\lambda_{y}+2G_{y}\right)\frac{\partial u_{y}}{\partial y}+G_{z}\frac{\partial u_{y}}{\partial z}+f_{y}=0, \end{array}$$
(59)$$\begin{array}{l} k_{3}\frac{\partial^{2}u_{x}}{\partial x\partial z}+\lambda_{z}\frac{\partial u_{x}}{\partial x}+G_{x}\frac{\partial u_{x}}{\partial z}+k_{3}\frac{\partial^{2}u_{y}}{\partial y\partial z}+\lambda_{z}\frac{\partial u_{y}}{\partial y}+G_{y}\frac{\partial u_{y}}{\partial z}\\ +k_{2}\frac{\partial^{2}u_{z}}{\partial x^{2}}+k_{2}\frac{\partial^{2}u_{z}}{\partial y^{2}}+k_{1}\frac{\partial^{2}u_{z}}{\partial z^{2}}+G_{x}\frac{\partial u_{z}}{\partial x}+G_{y}\frac{\partial u_{z}}{\partial y}+\left(\lambda_{z}+2G_{z}\right)\frac{\partial u_{z}}{\partial z}+f_{z}=0, \end{array}$$
where
(60)$$k_{1}=\lambda +2G,\;k_{2}=G,\;k_{3}=\lambda +G,\;$$
(61)$$\lambda_{x}=\frac{\nu }{\left(1+\nu \right)\left(1-2\nu \right)}\frac{\partial E}{\partial x},\;\lambda_{y}=\frac{\nu }{\left(1+\nu \right)\left(1-2\nu \right)}\frac{\partial E}{\partial y},\;\lambda_{z}=\frac{\nu }{\left(1+\nu \right)\left(1-2\nu \right)}\frac{\partial E}{\partial z},\;$$
(62)$$G_{x}=\frac{1}{2\left(1+\nu \right)}\frac{\partial E}{\partial x},\;G_{y}=\frac{1}{2\left(1+\nu \right)}\frac{\partial E}{\partial y},\;G_{z}=\frac{1}{2\left(1+\nu \right)}\frac{\partial E}{\partial z}.\;$$

### 4.2. Boundary Conditions

The displacement condition and traction condition are defined on boundaries $\Gamma_{1}$ and $\Gamma_{2}$ as
(63)$$u_{x}={\bar{u}}_{x},\;u_{y}={\bar{u}}_{y},\;u_{z}={\bar{u}}_{z},\;\left(x,y,z\right)\in \Gamma_{1},$$
(64)$$t_{x}=n_{x}\sigma_{xx}+n_{y}\sigma_{xy}+n_{z}\sigma_{xz},\;\left(x,y,z\right)\in \Gamma_{2},$$
(65)$$t_{y}=n_{x}\sigma_{xy}+n_{y}\sigma_{yy}+n_{z}\sigma_{yz},\;\left(x,y,z\right)\in \Gamma_{2},$$
(66)$$t_{z}=n_{x}\sigma_{xz}+n_{y}\sigma_{yz}+n_{z}\sigma_{zz},\;\left(x,y,z\right)\in \Gamma_{2},$$
where ${\bar{u}}_{x}$, ${\bar{u}}_{y}$, and ${\bar{u}}_{z}$ are given displacements; $n=\left[n_{x},n_{y},n_{z}\right]$ is the outward unit normal vector; and $\Gamma_{1}$ and $\Gamma_{2}$ indicate the Dirichlet boundary and the Neumann boundary, respectively.

### 4.3. Numerical Discretization

In this section, the governing equations of FGMs are discretized using the FBML and the FBMC. For the FBML, by substituting $N^{3}$ collocation nodes into partial derivatives, Equations (57)–(59) can be formulated in matrix form as
(67)$$\begin{array}{l} \left[k_{1}D_{x}D_{x}+k_{2}D_{y}D_{y}+k_{2}D_{z}D_{z}+\left(\lambda_{x}+2G_{x}\right)D_{x}+G_{y}D_{y}+G_{z}D_{z}\right]u_{x}\\ +\left(k_{3}D_{x}D_{y}+G_{y}D_{x}+\lambda_{x}D_{y}\right)u_{y}\\ +\left(k_{3}D_{x}D_{z}+G_{z}D_{x}+\lambda_{x}D_{z}\right)u_{z}+f_{x}=0, \end{array}$$
(68)$$\begin{array}{l} \left(k_{3}D_{x}D_{y}+\lambda_{y}D_{x}+G_{x}D_{y}\right)u_{x}\\ +\left[k_{2}D_{x}D_{x}+k_{1}D_{y}D_{y}+k_{2}D_{z}D_{z}+G_{x}D_{x}+\left(\lambda_{y}+2G_{y}\right)D_{y}+G_{z}D_{z}\right]u_{y}\\ +\left(k_{3}D_{y}D_{z}+G_{z}D_{y}+\lambda_{y}D_{z}\right)u_{z}+f_{y}=0, \end{array}$$
(69)$$\begin{array}{l} \left(k_{3}D_{x}D_{z}+\lambda_{z}D_{x}+G_{x}D_{z}\right)u_{x}\\ +\left(k_{3}D_{y}D_{z}+\lambda_{z}D_{y}+G_{y}D_{z}\right)u_{y}\\ +\left[k_{2}D_{x}D_{x}+k_{2}D_{y}D_{y}+k_{1}D_{z}D_{z}+G_{x}D_{x}+G_{z}D_{y}+\left(\lambda_{z}+2G_{z}\right)D_{z}\right]u_{z}+f_{z}=0, \end{array}$$
where $u_{d}={\left[u_{d1},u_{d2},\ldots ,u_{dN^{3}}\right]}^{T}$, d=x,y,z, denotes the displacement component vector with dimensions of $N^{3}\times 1$; $f_{d}={\left[f_{d1},f_{d2},\ldots ,f_{dN^{3}}\right]}^{T}$ is the body force component vector with dimensions of $N^{3}\times 1$; and $k_{\gamma }$, $\lambda_{d}$, and $G_{d}$ denote the diagonal coefficient matrix, expressed as
(70)$$k_{\gamma }=\left[\begin{array}{l} k_{\gamma }^{\left(1\right)}\;\;0\;\;\;\cdots \;\;0\\ \;\;0\;\;\;k_{\gamma }^{\left(2\right)}\;\;\;\cdots \;\;0\\ \;\cdots \;\;\;\;\cdots \;\;\;\;\cdots \;\;\;\;\cdots \\ \;\;0\;\;0\;\;\;\cdots \;\;\;\;k_{\gamma }^{\left(N^{3}\right)} \end{array}\right],\;\gamma =1,2,3,$$
(71)$$\lambda_{d}=\left[\begin{array}{l} \lambda_{d}^{\left(1\right)}\;\;0\;\;\;\cdots \;\;0\\ \;\;0\;\;\;\lambda_{d}^{\left(2\right)}\;\;\;\cdots \;\;0\\ \;\cdots \;\;\;\;\cdots \;\;\;\;\cdots \;\;\;\;\cdots \\ \;\;0\;\;0\;\;\;\cdots \;\;\;\;\lambda_{d}^{\left(N^{3}\right)} \end{array}\right],$$
(72)$$G_{d}=\left[\begin{array}{l} G_{d}^{\left(1\right)}\;\;0\;\;\;\cdots \;\;0\\ \;\;0\;\;\;G_{d}^{\left(2\right)}\;\;\;\cdots \;\;0\\ \;\cdots \;\;\;\;\cdots \;\;\;\;\cdots \;\;\;\;\cdots \\ \;\;0\;\;0\;\;\;\cdots \;\;\;\;G_{d}^{\left(N^{3}\right)} \end{array}\right].$$

The displacement components can be obtained directly by solving the linear system of Equations (67)–(69). For the FBMC, the matrix form of the governing equations can be expressed as
(73)$$\begin{array}{l} \left[k_{1}\Theta_{xx}+k_{2}\Theta_{yy}+k_{2}\Theta_{zz}+\left(\lambda_{x}+2G_{x}\right)\Theta_{x}+G_{y}\Theta_{y}+G_{z}\Theta_{z}\right]\alpha^{\left(u_{x}\right)}\\ +\left(k_{3}\Theta_{xy}+G_{y}\Theta_{x}+\lambda_{x}\Theta_{y}\right)\alpha^{\left(u_{y}\right)}\\ +\left(k_{3}\Theta_{xz}+G_{z}\Theta_{x}+\lambda_{x}D_{z}\right)\Theta_{z}\alpha^{\left(u_{z}\right)}+f_{x}=0, \end{array}$$
(74)$$\begin{array}{l} \left(k_{3}\Theta_{xy}+\lambda_{y}\Theta_{x}+G_{x}\Theta_{y}\right)\alpha^{\left(u_{x}\right)}\\ +\left[k_{2}\Theta_{xx}+k_{1}\Theta_{yy}+k_{2}\Theta_{zz}+G_{x}\Theta_{x}+\left(\lambda_{y}+2G_{y}\right)\Theta_{y}+G_{z}\Theta_{z}\right]\alpha^{\left(u_{y}\right)}\\ +\left(k_{3}\Theta_{yz}+G_{z}\Theta_{y}+\lambda_{y}\Theta_{z}\right)\alpha^{\left(u_{z}\right)}+f_{y}=0, \end{array}$$
(75)$$\begin{array}{l} \left(k_{3}\Theta_{xz}+\lambda_{z}\Theta_{x}+G_{x}\Theta_{z}\right)\alpha^{\left(u_{x}\right)}\\ +\left(k_{3}\Theta_{yz}+\lambda_{z}\Theta_{y}+G_{y}\Theta_{z}\right)\alpha^{\left(u_{y}\right)}\\ +\left[k_{2}\Theta_{xx}+k_{2}\Theta_{yy}+k_{1}\Theta_{zz}+G_{x}\Theta_{x}+G_{z}\Theta_{y}+\left(\lambda_{z}+2G_{z}\right)\Theta_{z}\right]\alpha^{\left(u_{z}\right)}+f_{z}=0, \end{array}$$
in which $\alpha^{\left(u_{d}\right)}={\left[\alpha_{1}^{\left(u_{d}\right)},\alpha_{2}^{\left(u_{d}\right)},\ldots ,\alpha_{N^{3}}^{\left(u_{d}\right)}\right]}^{T}$, d=x,y,z, denotes the unknown coefficient vector. Once the coefficient vectors are determined, the displacement components can be acquired from Equation (33).

## 5. Numerical Examples and Discussion

According to the COD criteria, once the displacement components on the crack surface are determined using the linear algebraic Equations (67)–(69) or Equations (73)–(75), the approximated SIFs can be evaluated directly. The relationships between the SIFs and the displacement components near the crack-tip are defined as
(76)$$\left\{\begin{array}{l} K_{I}\\ K_{\mathrm{II}}\\ K_{\mathrm{III}} \end{array}\right\}=\frac{E}{8\left(1-\nu^{2}\right)}\underset{r\to 0}{\lim}\sqrt{\frac{2\pi }{r}}\left\{\begin{array}{l} \Delta u_{z\prime }\left(r\right)\\ \Delta u_{x\prime }\left(r\right)\\ \left(1-\nu \right)\Delta u_{y\prime }\left(r\right) \end{array}\right\},$$
where $K_{I}$, $K_{\mathrm{II}}$, and $K_{\mathrm{III}}$ are the mode I, mode II, and mode III stress intensity factors, and *r* is the small distance from the crack front to the node on the crack surface, respectively. $\Delta u_{x\prime }$, $\Delta u_{y\prime }$, and $\Delta u_{z\prime }$ are the CODs in the local coordinate system, which is located at the front of the crack.

### 5.1. Central-Crack Plate under Tension Load

In this example, consider a central-crack plate for which T/W=1 and H/W=1 subjected to unit uniform loads $\sigma_{0}$ in the *z*-direction on the top boundary, as shown in Figure 2a.

The rigid constraint is imposed on the bottom boundary. The crack length is 2*a*, and the Poisson’s ratio was set as 0.3. For the normalized Young’s modulus *E*, consider the following three cases:Case 1—Homogeneous material: E=1.Case 2—Material gradation in the *y*-direction: $E=E\left(y\right)=E_{1}e^{\frac{y}{T}\ln\frac{E_{2}}{E_{1}}}$.Case 3—Material gradation in the *z*-direction: $E=E\left(z\right)=E_{1}e^{\frac{z}{H}\ln\frac{E_{2}}{E_{1}}}$.

$E_{2}$ and $E_{1}$ are the normalized Young’s modulus values at *y* = *T* (or *z* = *H*) and *y* = 0 (or *z* = 0), respectively. The ratio of the Young’s modulus is $E_{1}/E_{2}=2$. Due to the symmetry, only half of the plate is analyzed in this example. In the numerical procedure, for the FBM, the computational domain is divided into four blocks, as depicted in Figure 2b. The 15×15×15 roots of the Chebyshev polynomial of the first kind are distributed in each block. For the FEM, 63,600 20-node quadratic hexahedral elements and 15-node wedge elements with a total 269,377 nodes are used, as shown in Figure 3, while a UMAT subroutine was developed for FGMs. By utilizing the COD criteria, the results of the comparison of the normalized mode I SIF with respect to a/W=0.25, 0.5, 0.75 are given in Figure 4; in this section, we select the displacements of the third point in front of the crack tip for the SIF calculation in the FBM and FEM. Obviously, the numerical results of the FBML, the FBMC, and the FEM show very satisfactory agreement both in homogeneous material and FGM. In Figure 4, it can be observed that the SIFs of the homogeneous material are nearly uniform throughout most of the crack front. There are some differences in the SIFs at the two ends of the crack front and in the interior of the crack front due to the near-plain stress conditions. In addition, a difference in the SIF with a different number of collocation points was observed. This shows that convergent results pertaining to the SIF can be obtained when the number of collocation points along each side N>11. Furthermore, the normalized mode I SIF becomes larger with the increase in crack length. A comparison of the results also implies that the FBML and the FBMC are almost equivalent.

### 5.2. Edge-Crack Plate under Tension Load

An edge-crack plate with a fixed bottom subjected to unit uniform tensile loads $\sigma_{0}$ on the top boundary is discussed in the current example, as shown in Figure 5a. The scale parameters of the plate are T/W=2 and H/W=2. The crack length is *a* = 0.5, and the cracks form an angle θ with the *x* axis. The Poisson’s ratio is 0.3. Herein, the following three different situations are still considered.

Case 1—Homogeneous material: E=1.Case 2—Material gradation in *y*-direction: $E=E\left(y\right)=E_{1}e^{\frac{y}{T}\ln\frac{E_{2}}{E_{1}}}$.Case 3—Material gradation in *z*-direction: $E=E\left(z\right)=E_{1}e^{\frac{z}{H}\ln\frac{E_{2}}{E_{1}}}$.

$E_{2}$ and $E_{1}$ represent the normalized Young’s modulus at *y* = *T* (or *z* = *H*) and *y* = 0 (or *z* = 0), and assume that $E_{1}/E_{2}=2$. The local coordinate system x′y′z′ located on the crack surface is used to evaluate the CODs, namely, $\Delta u_{x\prime }$, $\Delta u_{y\prime }$, and $\Delta u_{z\prime }$, as depicted in Figure 5b.

In the computational process, for the FBM, the computational domain is divided into four blocks, as depicted in Figure 5b. The 15×15×15 roots of the Chebyshev polynomial of the first kind are distributed in each block. In ABAQUS with the UMAT subroutine, 67,360 20-node quadratic hexahedral elements and 15-node wedge elements with a total of 284,699 nodes were employed, as shown in Figure 6. Hence, we adopted the displacements of the third collocation node in front of the crack front to carry out an evaluation of the COD criterion; the comparison of the results of the normalized mode I and mode II SIFs with respect to $\theta =0^{\circ },\;30^{\circ },\;\;45^{\circ }$ are shown in Figure 7, Figure 8 and Figure 9. It is clear that the normalized mixed-mode SIFs of the FBML, the FBMC, and the FEM yield excellent consistencies in both homogeneous material and FGM.

In Figure 7, it can be observed that the SIFs of the homogeneous material have a nearly uniform distribution along most of the crack front with respect to $\theta =0^{\circ },30^{\circ },\;45^{\circ }$. Actually, the mode II SIFs are nearly zero due to the mode I edge-crack when $\theta =0^{\circ }$. And when $\theta \neq 0^{\circ }$, this edge-crack model transforms into a mixed-mode problem. The influence of material gradation along the *z*-direction is very small due to the isotropy on the crack surface. Furthermore, the tilt angle θ of the crack surface directly affects the distribution of mixed-mode SIFs. Similar to the central-crack, the normalized SIFs become larger with the increase in the tilt angle. The comparison of the results suggests that the FBML and the FBMC can achieve very high accuracy in three-dimensional edge-crack analysis.

### 5.3. Curved Edge-Crack under Shearing Load

In this example, consider a curved edge-crack plate with a fixed bottom subjected to unit uniform shearing load $\tau_{0}$ on the top boundary, which is shown in Figure 10a. The crack front is a curved line whose geometric parameters are shown in Figure 10b. The front end of the crack surface is an arc of radius *r* with its center at point *O*, where $r=\sqrt{{\left(0.5T\right)}^{2}+a^{2}}$ and *O* is the midpoint of line segment *AB, a* is the crack length. And we have T/W=2 and H/W=1. The Poisson’s ratio adopted is 0.3. Herein, the material gradation in the *y*-direction is considered, and the normalized Young’s modulus is $E=E_{1}e^{y/Tln\left(E_{2}/E_{1}\right)}$, in which $E_{1}/E_{2}=2$. $E_{2}$ and $E_{1}$ are the normalized Young’s modulus values at *y* = *T* and *y* = 0, respectively.

In the numerical computation, for the FBM, the computational domain is divided into four blocks, as depicted in Figure 10c. The 15×15×15 roots of the Chebyshev polynomial of the first kind are used in each block. In ABAQUS with the UMAT subroutine, as shown in Figure 11, 20-node quadratic hexahedral elements and 15-node wedge elements are employed, with a total number of 30,080 elements and 129,523 nodes. Then, the displacements of the third point in front of the crack tip are selected to calculate the SIFs; the comparison of the results of the normalized mode I, mode II, and mode III SIFs with respect to a/W=0.25, 0.5, 0.75 are depicted in Figure 12. There is little difference in the numerical results regarding the SIFs of the FBML, the FBMC and the FEM. This further validates that the FBML and the FBMC are equivalent numerically.

In Figure 12, it can be observed that the related error among the solutions of the FBML, the FBMC, and the FEM is approximately less than 10%. Compared with the straight crack front, the fluctuation of the SIFs of the curved crack is larger. And the mode I and mode II SIFs possess a zero-point due to the distribution of displacements. In addition, the mode III SIFs increase with the increase in crack length a/W. Through comparisons of the SIFs presented above, the FBML and the FBMC can achieve high accuracy in the analysis of three-dimensional mixed-mode fracture mechanics.

## 6. Conclusions

A finite block method based on Lagrange polynomial interpolation (FBML) and Chebyshev polynomial interpolation (FBMC) for three-dimensional static linear elastic fracture mechanics was developed in this study. The numerical algorithm was validated using three typical examples. The results obtained using ABAQUS software 2020 with the UMAT subroutine were taken as a benchmark for comparisons. The following conclusions were drawn: (1) the FBM can achieve very high accuracy with a few blocks when addressing problems in three-dimensional linear elastic fracture mechanics; (2) although the methods of discretization are different, the results of the FBML and the FBMC are almost equivalent; (3) it is not necessary to refine collocation nodes near the crack tip when using the strong-form FBM; and (4) the FBM can be extended to more complicated problems, such as multi-physics, plastic mechanics, and three-dimensional contact problems.

## Figures and Tables

**Figure 1 materials-16-07301-f001:**
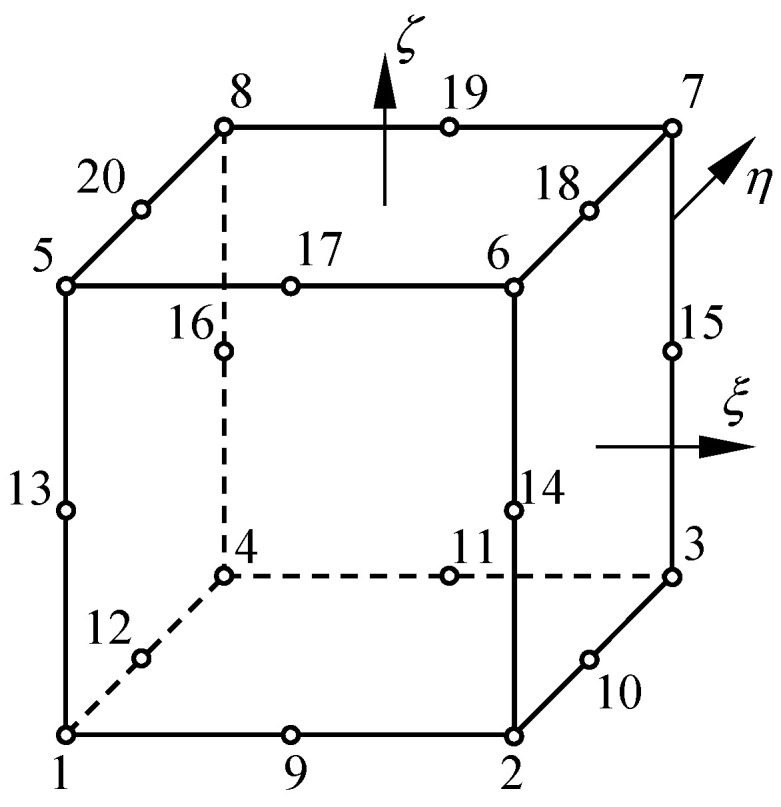
Three-dimensional normalized domain and its seeds.

**Figure 2 materials-16-07301-f002:**
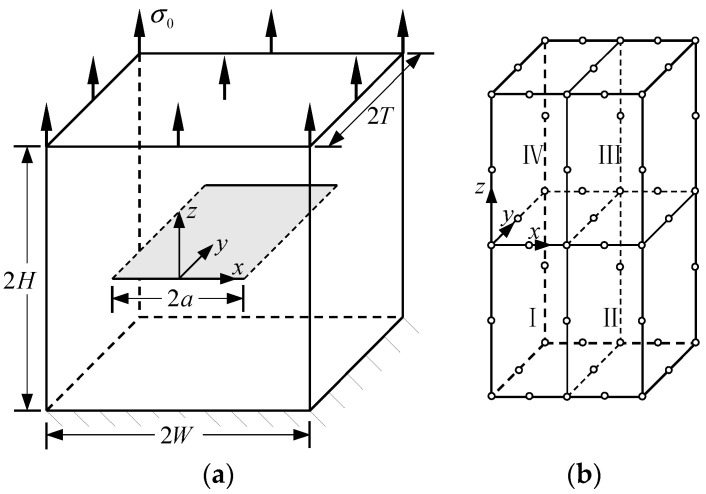
Central-cracked plate: (**a**) geometry of the plate; (**b**) half of the plate divided by four blocks.

**Figure 3 materials-16-07301-f003:**
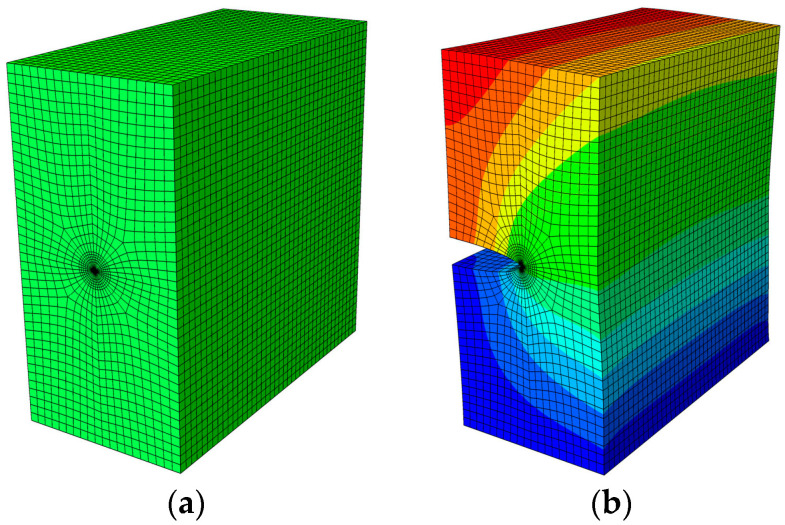
The FEM model: (**a**) mesh distribution of the half plate; (**b**) results regarding $u_{z}$ of the half of the plate.

**Figure 4 materials-16-07301-f004:**
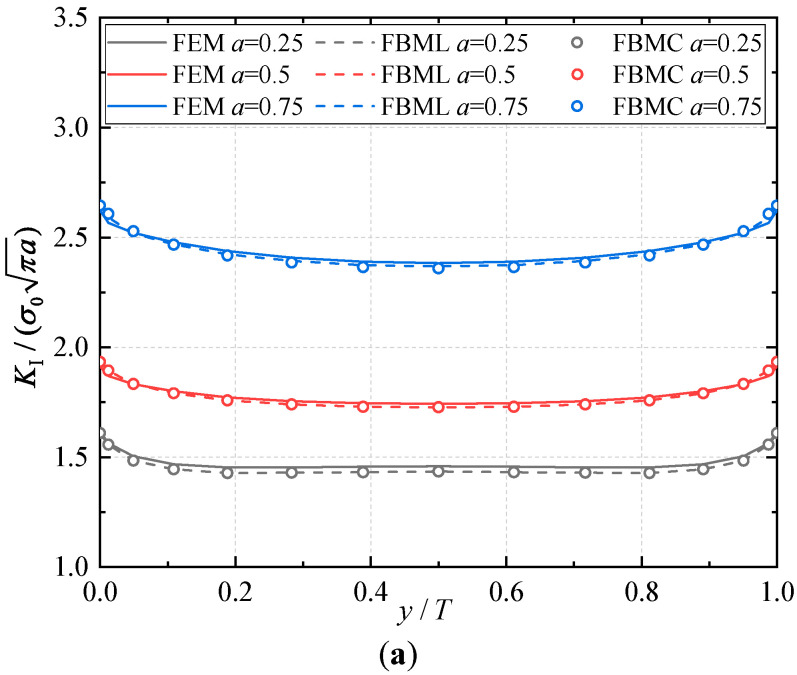
Comparison of the results obtained for the FEM, FBML, and FBMC: (**a**) normalized mode I SIFs of homogeneous material; (**b**) normalized mode I SIFs of material gradation in *y*-direction; (**c**) normalized mode I SIFs of material gradation in *z*-direction.

**Figure 5 materials-16-07301-f005:**
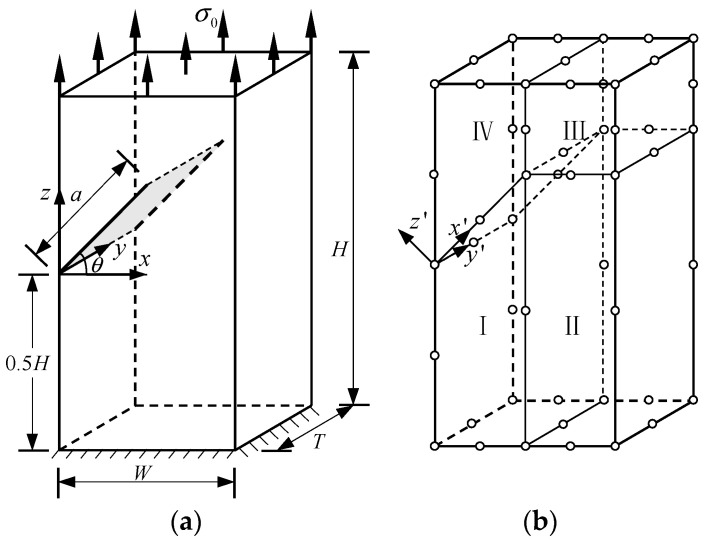
Edge-crack plate: (**a**) geometry of the plate; (**b**) the whole plate divided by four blocks.

**Figure 6 materials-16-07301-f006:**
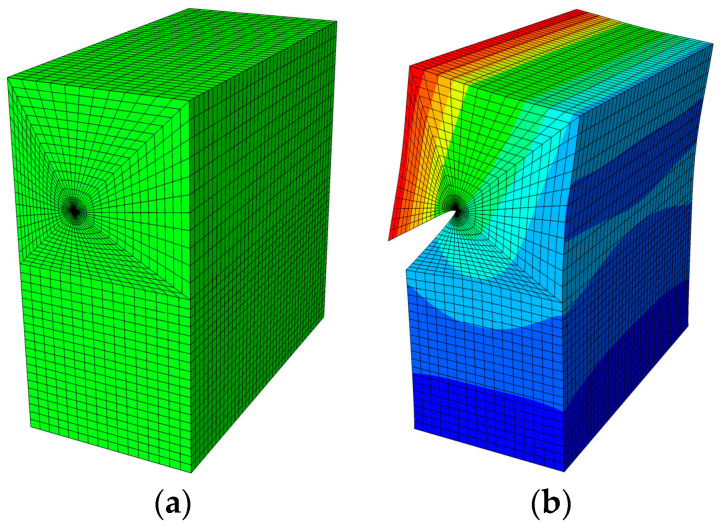
The FEM model: (**a**) mesh distribution of the whole plate with a slant edge crack; (**b**) results regarding $u_{z}$ of the whole plate.

**Figure 7 materials-16-07301-f007:**
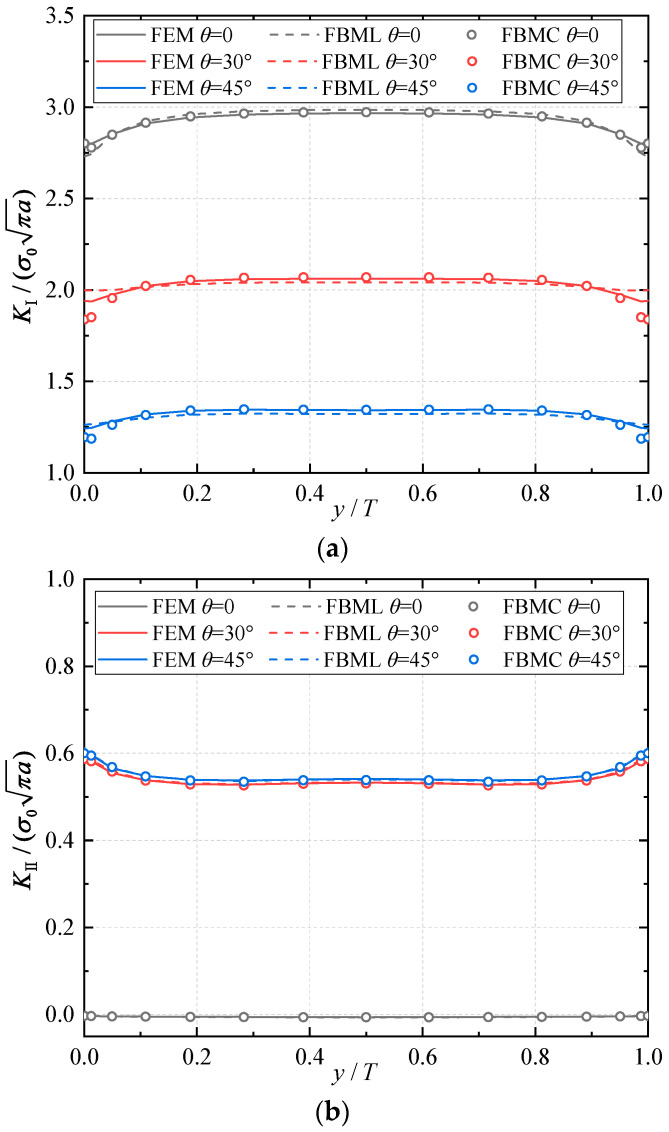
Result comparisons for homogeneous material: (**a**) normalized mode I SIFs; (**b**) normalized mode II SIFs.

**Figure 8 materials-16-07301-f008:**
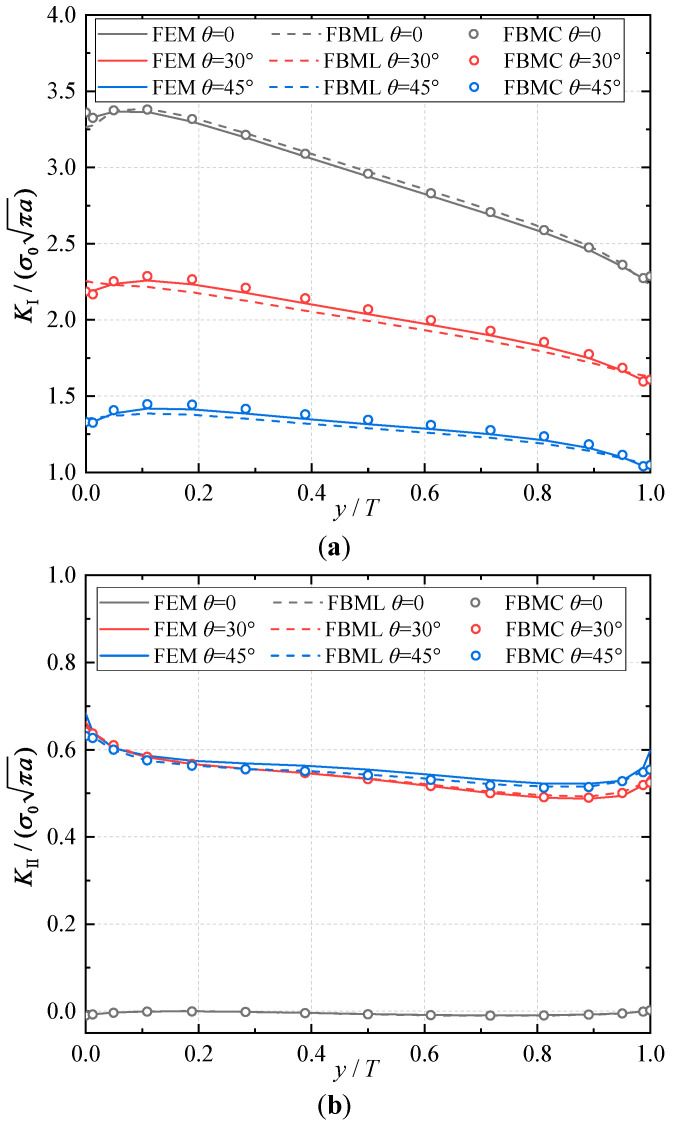
Result comparisons for FGM with material gradation in *y*-direction: (**a**) normalized mode I SIFs; (**b**) normalized mode II SIFs.

**Figure 9 materials-16-07301-f009:**
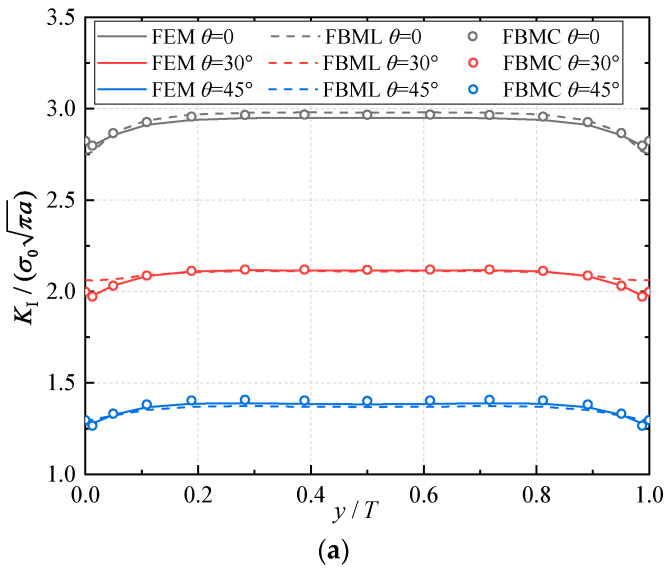
Result comparisons for FGM with material gradation in *z*-direction: (**a**) normalized mode I SIFs; (**b**) normalized mode II SIFs.

**Figure 10 materials-16-07301-f010:**
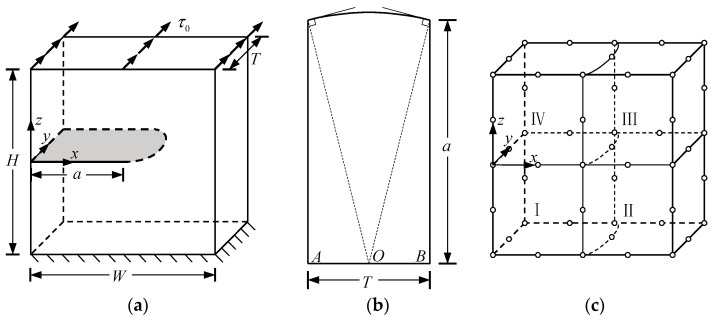
Curved edge-crack plate: (**a**) geometry of the plate; (**b**) geometry of crack front; (**c**) the whole plate divided by four blocks.

**Figure 11 materials-16-07301-f011:**
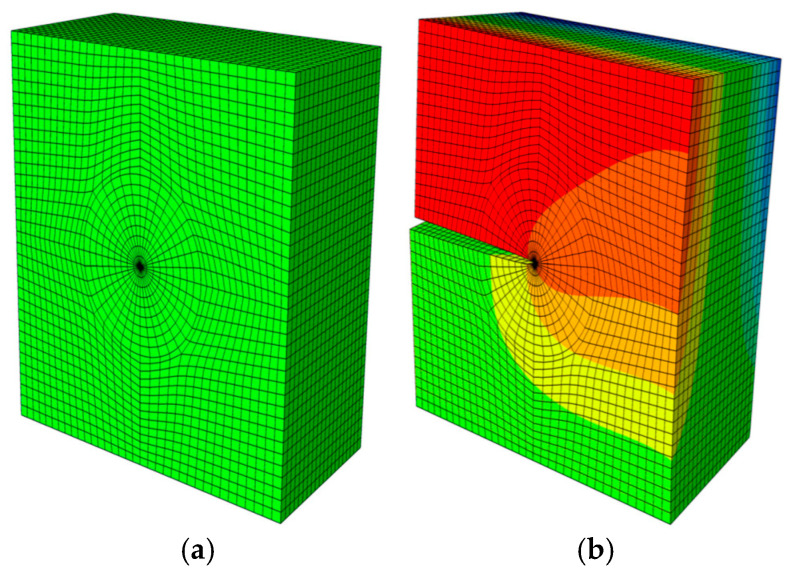
The FEM model: (**a**) mesh distribution of the whole plate with a curved edge crack; (**b**) results regarding $u_{z}$ of the whole plate.

**Figure 12 materials-16-07301-f012:**
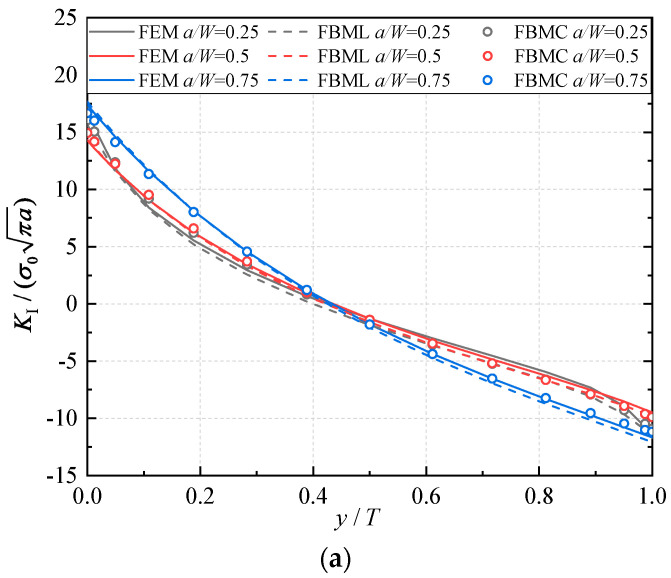
Result comparisons: (**a**) normalized mode I SIFs; (**b**) normalized mode II SIFs; (**c**) normalized mode III SIFs.

## Data Availability

The data presented in this study are available on request from the corresponding author.

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
