# Peer review of "Three-Dimensional Fracture Analysis in Functionally Graded Materials Using the Finite Block Method in Strong Form"

_materials, 2023, doi:10.3390/ma16237301_

Round 1

Reviewer 1 Report

Comments and Suggestions for Authors

The authors present a compelling approach for utilizing the strong form finite block method (FBM) in three-dimensional fracture analysis involving functionally graded materials. This study is both timely and valuable. The findings are effectively structured and presented in a suitable manner. Additionally, the reviewer agrees with the organization and format of the manuscript. Nevertheless, it is essential to underscore and emphasize the novelty of the present research effort. Equations 10 and 11 are absent from the provided content. Some equations are presented in bold font, while others are not. It is important for authors to compare their work to that which has been previously published.

Comments on the Quality of English Language

Minor editing of English language required

Author Response

See file attached.

Reviewer 2 Report

Comments and Suggestions for Authors

In the manuscript entitled “Strong form finite block method for three-dimensional fracture analysis in functionally graded materials”, the applications of finite block method for three-dimensional fracture analysis in functionally graded materials are presented. Several numerical examples are demonstrated.

The manuscript is well-written and has a good structure. On the review opinion, the manuscript can be considered for publication after addressing some minor revisions.

1. In the reviewer's opinion, the title of the paper mentioning the strong form of the method appears somewhat confusing. The concern arises from the limited extent of the explanation of the strong form within the content of this paper.

2. p.10. It is unclear to which case (Case 1, Case 2, or Case 3) Figure 3 is associated.

3. p.10 Providing additional details on the implementation of the subroutine UMAT for the considered functionally graded body is strongly recommended.

4. p.10 Could you provide a more detailed discussion on the implementation of the COD criterion in the calculation of SIFs? At what distance from the crack front have the displacements been computed?

5. Besides mode I in Section 5.1, do modes II and III exist in the case of FGM materials ?

6. Can you provide information on whether singular elements were used in the calculations?

7. The same questions 1), 2), 5) might be applied to Section 5.2.

8. The same questions 1), 2), 5) might be applied to Section 5.3.

9. p.15 Could you share details on the modeling of a curved crack front in Section 5.3?

10. The reviewer only would like to draw attention to the fact that one of the corresponding authors is cited in 21 from of the 44 references presented in the paper. 

Author Response

See the file attached.

Reviewer 3 Report

Comments and Suggestions for Authors

The paper presents the so-called finite block method (FBM) to 3D fracture analysis with functionally graded materials. In principle, authors develop a 1D differential matrix and extend it to 3D crack analysis problems by using Lagrange and Chebyshev interpolation polynomialsThe paper is well written and exceptionally summarized, nevertheless it can be ameliorated by addressing the following few points:

1- Authors claim they present a "strong form" of the FBM without clearly justifying its meaning. Actually, the question that arises in my mind whether there could exist weak or variational forms.

2- The method takes as a reference solutions given by the finite element method, and it is quite logic as closed-form solutions may not be available for all cases. Authors claim that 15 points for each side of each block is enough. I would like to see the "evolution" of getting the numerical solutions as the number of seeds increases, at least for the first example. An error vs. number-of-seeds graph could be sufficient.

3- Where does the material gradation take place? I guess it is in the definition of matrix K in (65). I guess it needs further. By the way, how is the approximation of the material inhomogeneiy performed? For example, in finite element method one can either assign to each element a constant material property or make the material property variable within the element by the very same shape functions used for the displacemnt field. Could be there an equivalent strategy in the FBM?

4- Authors may address other works where the idea of direct collocation method is employed for problems in elasticity, e.g., 10.1016/j.compstruct.2023.116784 and 10.1007/s00466-005-0008-7

Author Response

See the file attached.

Round 2

Reviewer 3 Report

Comments and Suggestions for Authors

Authors replied to the raised points clearly. I recommend the publication of the manuscript in it's current form.